# DNA–Lysozyme Nanoarchitectonics: Quantitative Investigation on Charge Inversion and Compaction

**DOI:** 10.3390/polym14071377

**Published:** 2022-03-28

**Authors:** Rongyan Zhang, Yanwei Wang, Guangcan Yang

**Affiliations:** Department of Physics, Wenzhou University, Wenzhou 325035, China; zhangry0128@163.com

**Keywords:** lysozyme, DNA compaction, condensing forces

## Abstract

The interaction between DNA and proteins is fundamentally important not only for basic research in biology, but also for potential applications in nanotechnology. In the present study, the complexes formed by λ DNA and lysozyme in a dilute aqueous solution have been investigated using magnetic tweezers (MT), dynamic light scattering (DLS), and atomic force microscopy (AFM). We found that lysozyme induced DNA charge inversion by measuring its electrophoretic mobility by DLS. Lysozyme is very effective at neutralizing the positive charge of DNA, and its critical charge ration to induce charge inversion in solution is only 2.26. We infer that the high efficiency of charge neutralization is due to the highly positively charged (+8 e) and compact structure of lysozyme. When increasing the concentration of lysozymes from 6 ng·µL^−1^ to 70 ng·µL^−1^, DNA mobility (at fixed concentration of 2 ng·µL^−1^) increases from −2.8 to 1.5 (in unit of 10^−4^ cm^2^·V^−1^·S), implying that the effective charge of DNA switches its sign from negative to positive in the process. The corresponding condensing force increased from 0 pN to its maximal value of about 10.7 pN at concentrations of lysozyme at 25 ng·µL^−1^, then decreases gradually to 3.8 pN at 200 ng·µL^−1^. The maximal condensing force occurs at the complete DNA charge neutralization point. The corresponding morphology of DNA–lysozyme complex changes from loosely extensible chains to compact globule, and finally to less compact flower-like structure due to the change of attached lysozyme particles as observed by AFM.

## 1. Introduction

The interactions between nucleic acids and proteins play an important role in basic intracellular processes such as transcription, translation, and cell division, which are essential for various functions of living organisms [1,2]. Meanwhile, they are also related with the liquid–liquid phase separation (LLPS) process of proteins and nucleic acids, which lead to the formation of membrane-less organelle in cells. LLPS plays a crucial role in various normal cellular activities, such as the assembly of stress granules, nucleoli, and the aggregation of proteins related to glucose metabolism and cell signaling. Lysozyme is an enzyme, commonly used to characterize protein folding and aggregation e.g., examining the effects of molecules on protein aggregation or amyloid formation [3,4,5,6]. Lysozyme undergoes LLPS in a solution containing a high concentration of salt or by mixing with cations [5]. In addition, lysozyme and histone are similar in structure, so it is also helpful to elucidate the interaction between lysozyme and DNA as a reference for analyzing chromatin [7,8]. Moreover, lysozyme is a “killer protein” against HIV due to its interaction with nucleic acids [9,10,11]. Therefore, in the present study, we studied the charge compensation and compaction effect of lysozyme on DNA to further elucidate the interaction between DNA and charged proteins.

Forming protein–DNA structure in solution is a complex dynamic process involving various forces including electrostatic, hydrophobic, and spatial interactions [12,13], which are also main derivation for various biomolecule nanoarchitectonics [14]. Usually, DNA carries a practically homogeneous high negative charge in solution due to its fully deprotonated phosphate backbone while proteins generally show a heterogeneous charge distribution with anionic, cationic, as well as hydrophobic patches. In addition, both the effective charge and the heterogeneous charge distribution of a protein can be regulated by the interactions with charged agents and/or solution conditions [15]. In a previous mechanical measurement of DNA–lysozyme complexes in optical tweezers [1], they found that the persistence length of DNA–lysozyme complexes exhibits a nonmonotonic behavior as a function of the protein concentration. We have studied DNA complexes at various ionic and solution conditions in the past [16,17,18,19,20,21,22,23,24,25].

Lysozyme is a common small molecular enzyme with antibacterial function. It has a compact structure with several helices surrounding a small beta sheet region. Under neutral solution conditions, lysozyme is positively charged. The interactions between DNA and lysozyme have been investigated by various techniques such as atomic force microscopy [26], optical microscopy [27], interferometry [28], small angle X-ray scattering, and light scattering technique [29]. It was suggested that electrostatic and/or hydrophobic interactions between lysozyme units are important factors driving phase separation in DNA–lysozyme systems [1,30]. There are strong indications that direct interactions between the protein units and the electrostatic attraction between DNA and lysozyme are instrumental in controlling the morphology of the formed assemblies [10,11,12]. Despite the encouraging advancements, the properties of DNA–lysozyme interactions have not been understood extensively, especially at the quantitative and single molecular levels. In the present study, we try to fill the gap and use magnetic tweezers (MT), atomic force microscopy (AFM), and dynamic light scattering (DLS) to quantitatively characterize the interaction between lysozyme and DNA.

## 2. Materials and Methods

### 2.1. Materials

Double-stranded λ-phage DNA (48,502 bp) in the present study was purchased from New England Biolabs Co. (Ipswich, MA, USA) and did not go through purification before use. TE buffer (10 mM Tris-HCl and 1 mM EDTA, pH = 8.0) was used for the stock solution of DNA, and the original concentration of was 500 ng·µL^−1^. The other biochemical agents (such as lysozyme, bovine serum albumin (BSA), and hydroxylmethylaminoethane (TRIS)) were purchased from Sigma-Aldrich (St. Louis, MO, USA)and used as received. The deionized water of resistivity 18.2 MΩ⋅m was purified through the Milli-Q system (Millipore Corporation, Burlington, MA, USA) and was used for all solution preparation. Stock solution containing 1 mg·mL^−1^ lysozyme was prepared in 1 mM Tris-HCL buffer (pH = 7). All chemical agents were used as received. We repeated all experiments at least twice to obtain consistent results.

### 2.2. Magnetic Tweezers Experiment

A transverse MT setup was built on an inverted microscope (Nikon TE2000U, Tokyo, Japan) for studying the dynamic process of tethering DNA, as shown in Figure 1A. The detailed description of MT setup was presented in previous works [18,23,31]. The procedure for force measurement can be briefly described as follows: one end of DNA molecule is linked to an immobile substrate (the glass sidewalls) and other end is attached to a paramagnetic bead [32]. When both the two ends of DNA were linked solidly together, we used a permanent magnet controlled by a micromanipulator system (MP-285, Novato, CA, USA) to exert force to the DNA molecule by approaching the paramagnetic bead and then pulling the tethered DNA chain. We used a CCD camera to record the movement of paramagnetic in real-time. The movie of the small magnetic spheres was captured by a Fast Fourier Transform-based software, and the video was analyzed by Mathcad software. We calculated the condensing force of DNA molecules by fitting the data to the worm-like-chain (WLC) model [22,33]. We define the critical condensing force (Fc) as the force when the first steplike contracting occurred in DNA extension-time plot.

The protocol to tether DNA can be described briefly as follows: We linked the ends of λ-DNA chemically labeled, single-stranded, 12-base oligonucleotides (3′-digoxigenin-tccagcggcgggand 3′-biotin-cccgccgctgga) [34]. Then 0.5 µL of a stock solution of streptavidin-coated magnetic beads (M-280, Dynal Biotech, Bromborough, UK) was gently incubated with 0.5 µL of modified DNA solution for 30 min in 150 µL buffer. After the incubation, we found DNA-bead structures formed.

We flushed the lysozyme solution at different concentrations into the sample cell by a syringe pump. At first, we had to find a single suspending λ-DNA and pulled the DNA to its maximal length (about 16 µm) by a force of more than 10 pN, which is achieved by moving the permanent magnet close to the paramagnetic bead. Then, we moved the magnet slowly away to lower the force to a needed value. Because of the presence of lysozyme in solution, the tethered DNA usually shrink when the applied force is lowered to a small enough value. We then monitor and record the conformational change of DNA. Figure 1B,C presents typical extension-time curves of DNA compaction induced by lysozyme, in which (B) corresponds to no DNA condensing case while (C) shows a condensing curve of DNA.

### 2.3. Electrophoretic Mobility Measurement (EM) and Size Measurement by DLS

Electrophoresis-mobility (EM, μ) measurement were accomplished by using a DLS setup of Malvern Zetasizer nano ZS90 (Malvern Instruments Limited Company, Malvern, UK) with M3-PALS technique. The light source of the DLS is a He–Ne gas laser (λ = 633 nm) [22,33]. We received the light scattering signal by an avalanche photodiode mounted on the goniometer arm in perpendicular to the direction of the incident light. We diluted the DNA samples to a concentration of 2 ng·µL^−1^ by adding a buffer solution of 1 mM Tris (pH = 7). Then we pipetted lysozyme to the solution to obtain different concentration of samples for measurement. Before starting a measurement, we incubated the mixed solution for 10 min at room temperature. We used 1 mL volume of DNA solution for measurement, and the temperature of sample cell was controlled at 25 °C.

In particle size measurements, we also diluted the DNA samples to a concentration of 2 ng·µL^−1^ with a buffer solution containing 1 mM Tris (pH = 7). Similarly, we incubated all samples for 10 min at room temperature. We took 50 µL volume of DNA solution in the measurement, and the temperature of sample cell was set to 25 °C.

### 2.4. Atomic Force Microscopy (AFM)

The detail description of sample preparation for AFM can be found in previous published work by us and other groups [17,25,35]. The process can be briefly described as follows: mica square (1 × 1 cm^2^) attached to glass slide preparing as substrates for DNA adsorption. We mixed DNA with the lysozyme solution. The DNA samples were also diluted to a concentration of 2 ng·µL^−1^ and the concentration of lysozyme ranged from 0–18 ng·µL^−1^. The complex solution was incubated for 30 min at room temperature. Then 30 µL of lysozyme–DNA complex solution was deposited onto the freshly cleaved mica surface and incubated for 3 min. After the incubation, the sample was then rinsed with purified water ten times and blow dried gently with nitrogen gas. The lysozyme–DNA complex was fixed onto the mica surface. We used AFM (JPK Instruments AG, Berlin, Germany) in AC mode to scan all the prepared samples to obtain images. In the scanning mode, we used a silicon AFM tip (NCHR-50, Nano World Corporation, Neuchatel, Switzerland) with aluminum coating. The spring coefficient and the resonance frequency of the tip are 42 N·m^−1^ and 320 kHz, respectively. We chose a 5 μm × 5 μm area for imaging and the scan rate is set to 1.0 Hz. The resolution of images obtained is 512 × 512 pixels (4–6 nm per pixel).

## 3. Results

### 3.1. Condensing Force of DNA–Lysozyme Complex

It is well-known that DNA can be condensed by many small cations (with valences equal or greater than 3) such as spermidine and Co(NH_3_)_6_^3+^. The process is always companied by DNA charge neutralization by the counterion [36]. When DNA is mixed with positively charged polyelectrolytes such as cationic polymers, polymerization formation or macrophase separation often occurs [21]. Since DNA is negatively charged in solution, the process of DNA compaction must overcome the electrostatic repulsion between segments of DNA so that one or few DNA chains can form a more compact and ordered structure. Thus, DNA compaction involves like-charge attraction, in which the attracting or condensing force appears to be related to its charge neutralization. In order to study the interaction between lysozyme and DNA quantitatively, we tried to pull the DNA–lysozyme complexes in a flow cell with MT, as described in the previous materials and methods section. In the setup, the tethered DNA may shrink under the influence of lysozyme and can be unraveled by moving the magnet, and the tethering or condensing force is measured simultaneously.

We measured the condensing force of DNA induced by lysozyme. Figure 2 shows the DNA contraction curve as a function of time under different lysozyme concentrations by MT. When the lysozyme concentration is low, such as 6 ng·µL^−^^1^ as shown in Figure 2A, we can see that, the length of DNA shrinks with time gradually and slowly, but there is no step-like contraction, implying that no DNA condensation occurred, and the corresponding condensing force was around 0 pN. When the concentration of lysozyme is continuously increased to 18 ng·µL^−^^1^, the DNA extension curve shows many small jumps in the shrinkage process, and the corresponding condensation force is about 7.1 pN, as shown in Figure 2B. Then we increased the solution concentration further to 25 ng·µL^−^^1^, as shown in Figure 2C, where the maximum critical cohesion of DNA is 10.7 pN. When the solution concentration was further increased, the F_C_ of condensation force gradually decreased, as shown in Figure 2D. If the concentration of lysozyme reaches 50 ng·µL^−^^1^, F_C_ of cohesion decreases to 8.2 pN. The force lowers further to 6.9 pN for 100 ng·µL^−^^1^ lysozyme concentration. Figure 2F summarizes the dependence of condensation force on lysozyme concentrations. As can be seen from the whole curve, with the increase of lysozyme concentration, the condensing force increases and reaches the largest value of about 10.7 pN at the concentration of lysozymes at 25 ng·µL^−^^1^. Then the condensing force gradually decreases to 3.8 pN, at a concentration of lysozyme of 200 ng·µL^−^^1^.

To explore pH effect on DNA/lysozyme interaction, we measured the condensing force of DNA at 25 ng·µL^−1^ lysozyme concentration under basic and acidic conditions. The result is shown in Figure 3. The measured condensing force is 10.9 pN at pH = 5.3, and becomes 6.7 pN at pH = 8.5. It implies that the structure of DNA–lysozyme complex becomes more compact in acidic solution.

### 3.2. DNA Electrophoresis Mobility and Size in Lysozyme Solutions

DNA is a negatively charged biological polyelectrolyte, so it has a negative zeta potential and electrophoretic mobility in an aqueous solution because of its protonated phosphate backbone. The electrophoretic mobility of DNA complex, µ, is proportional to its net charge, which is composed of the bare charge of DNA and the charge of counterions attracted to its surface [20,37]. It reflects the extent of charge neutralization of DNA by counterions in solution. In order to figure out the fraction of the charge neutralization of DNA by lysozyme, we measured the electrophoretic mobility (EM) of DNA at various lysozyme concentrations.

Figure 4 shows the electrophoretic mobility of DNA measured as a function of lysozyme concentration in a neutral solution of pH = 7. As we can see in the Figure 4, with the increase of lysozyme concentration, the electrophoretic mobility of DNA gradually increases from about −2.8 × 10^−4^ cm^2^·V^−1^·S (10^−4^ cm^2^·V^−1^·S is used for the unit of the electrophoretic mobility consistently below and will not be presented repeatedly for clarity) to eventually becoming positive. When the concentration of lysozyme reached 18 ng·µL^−1^, the EM of DNA complex becomes around −1.5. If the concentration increases further to about 25 ng·µL^−1^, the mobility is almost equal to 0. This means that the negative charge ionized by the DNA phosphate skeleton is neutralized completely. The corresponding critical charge ratio of DNA induced by lysozyme is only 2.26 for charge inversion, indicating that it is more effective than common cations such as spermine and polylysine K_8_. If we continue to increase the concentration of lysozyme in solution, the mobility of the complex switches from negative to positive, which corresponds to the case of overcompensation of DNA charge, in other words, charge inversion occurs. With the increase of lysozyme concentration, the electrophoretic mobility goes up gradually and finally reaches a saturated value. We can see that the electrophoretic mobility reached and stagnated at around 1.5 at a lysozyme concentration of 50 ng·µL^−1^. Generally, DNA charge neutralization is closely related with its condensation. If we compare the concentration dependence of condensing forces in Figure 2 with the present mobility of DNA complexes, it is easy to find that the complete charge neutralization corresponds to the maximal condensing force when lysozyme concentration is about 25 ng·µL^−1^. Both below and beyond this concentration, the condensing force becomes smaller because of the negative or positive electrostatic repulsion. We will explore the phenomenon further by measuring the DNA complex size and observe the morphology of the structure directly by AFM.

The pH effect is also reflected in the electrophoretic mobility of the complex, as shown in Figure 5. We can see that the needed lysozyme concentration for charge neutralization is less in acidic solution (18 ng·µL^−1^ at pH = 5.3) than in basic solution (25 ng·µL^−1^ at pH = 8.5). It implies that low pH promotes charge neutralization of DNA by lysozyme.

As is expected, charge neutralization significantly influences DNA compaction and coagulation because of Coulombic repulsion between the charged segments of DNA. In general, the more DNA charges are compensated or neutralized by counterions in the solution, the more compact the DNA conformation becomes. We used DLS to measure particle sizes of DNA at different concentrations of lysozyme, as shown in Figure 6. We can see that the size of lysozyme–DNA complex decreases at first and then increases with lysozyme concentration. When lysozyme is absent in the solution, the size of DNA is around 500 nm. When 2 ng·µL^−1^ lysozyme was added in, the size of the lysozyme–DNA complex decreased to about 350 nm. If lysozyme concentration is increased to 25 ng·µL^−1^, the size of the complex reached its minimal value of 150 nm. After that, the particle size goes up gradually. For example, when we increased lysozyme concentration to 35 ng·µL^−^^1^, the size of the complex becomes 170 nm, and it continues to increase in the subsequent process. If the concentration of lysozyme is 70 ng·µL^−1^, the particle size is about 250 nm, which is about the same size at 5 ng·µL^−1^ lysozyme concentration.

The form factor of DNA is closely related with its charge neutralization by lysozyme through electrostatic interaction. From Figure 4 and Figure 6, we can see that the size of lysozyme–DNA complex decreases initially with increasing concentration of lysozyme since the charge of DNA is neutralized. When the concentration of lysozyme reached 25 ng·µL^−1^, the electrophoretic mobility showed that the negative charge on DNA was almost completely neutralized and positively charged, that is, the lysozyme–DNA complex was positively charged as a whole, but due to the Coulomb repulsion between positive charges, DNA became loose and the particle size of the complex became larger. On the other hand, due to the interaction between positively charged residues and negatively charged residues on lysozyme, the number of proteins on the complex also increased, which further led to the increase of the particle size of the complex.

### 3.3. AFM Morphology of DNA

To observe the effect of lysozyme on DNA morphology more directly, we used AFM to capture the images of DNA chains in presence of different concentrations of lysozyme. The interaction between DNA and lysozyme has been studied by AFM in a slightly different way [10]. The researchers developed a simple method for assembling protein lysozyme onto the plasmid DNA which has been deposited on the APTES-modified mica surface. The density of lysozyme molecules assembled on the substrate DNA can be adjusted by changing the protein concentration.

As described in the experiments and materials methods section, the mixed solution of DNA and lysozyme was incubated at room temperature for 0.5 h before AFM imaging. Then the solution was pipetted to a fresh mica disk surface for its adsorption and drying by nitrogen gas blowing. After these steps, the mica disk was mounted to AFM for scanning to obtain an image. Figure 7 shows DNA morphologies at different concentrations in a solution of lysozyme. In previous studies, it was known that DNA molecules were condensed or compressed by divalent counterions. Therefore, the morphology of DNA in solution containing divalent magnesium ions was imaged for comparison, as shown in Figure 7A. When the concentration of lysozyme is 6 ng·µL^−1^, as shown in Figure 7B, we can see flower-liked DNA and many lysozymes bind to the DNA skeleton. When we further increased the concentration of the solution, the DNA conformation changes to dense bead string structures and the lysozymes spread all over the DNA, as shown in Figure 7C. When C = 18 ng·µL^−1^, more lysozymes bind the DNA bead string or lysozymes bind lysozymes to form a dimer and the DNA conformation condensed to a globule shape in Figure 7D. When the concentration of lysozyme was further increased to 35 ng·µL^−1^ (Figure 7E) and 50 ng·µL^−1^ (Figure 7F), the DNA conformation remained unchanged. It is obvious from the figure that as more and more DNA condensed, more and more lysozyme proteins were attached. From these AFM images, we can see the binding of lysozyme to DNA as shown in Figure 7A–F.

When lysozyme is attached to DNA, one concern is whether its conformation can be maintained. We used Fourier transformed infrared spectroscopy to characterize the conformation change of lysozyme. As shown in Figure 8, we can see that infrared absorbance of lysozyme solution with and without DNA shows very little difference of its amide I characteristic peak [38]. Thus, we assume that lysozyme does not undergo conformation change when combining with DNA.

## 4. Discussion and Conclusions

We have measured the mechanical and electrokinetic properties of DNA–lysozyme complex in previous sections. In this study, we assumed that the interaction between lysozyme and DNA was spontaneous and irreversible. These specific features can be ascribed to the strong electropositivity of lysozyme in neutral solution (pH = 7). When the concentration of lysozyme is low in the bulk solution, only a small part of the negative charge of DNA phosphate skeleton is neutralized, so the electrostatic repulsion is still quite strong between the DNA segments, accompanying no DNA compaction. This corresponds to the case shown in Figure 1B. In the Figure, the DNA extension curve appears as an almost horizontal downslope with no shrinking steps. In this case, the corresponding condensing force of DNA is virtually 0 pN. As the concentration of lysozyme increases, the negative charge of DNA–lysozyme complex decreases, and the coulombic repulsive barrier is weakened. Therefore, DNA segments become attractive to each other due to the weakened electrostatic repulsion and conformational entropic effect, leading to DNA compaction as shown in Figure 1C. If we continue to increase the concentration of lysozyme, the negative charge of DNA in the solution is entirely neutralized by counterions. Furthermore, DNA can attract more positive charge than its own negative charge, implying the whole DNA–lysozyme complex becomes positively charged, i.e., charge inversion occurred. As shown in the schematic cartoons in Figure 6, the charge neutralization is a key factor for DNA compaction by lysozyme to overcome the electrostatic repulsion between segments of DNA chain. In the process of charge neutralization and overcompensation, the morphology of DNA changes from loosely spread to compact rods or globules, and finally to flower-like shapes containing a compact core. Correspondingly, with increasing lysozyme concentration in solution, the particle size of the DNA complex measured by DLS goes down initially, then reaches the minimal value. After that point, it goes up gradually due to the charge inversion.

In summary, we studied the interaction between DNA and lysozyme, and explored the effects of lysozyme concentration on DNA compaction. The concentration of lysozyme influences DNA charge neutralization and compaction significantly. We found DNA charge inversion occurred when the charge ratio of DNA to lysozyme is about 2.26, which is much less than the corresponding values of poly-lysine K_8_ (12.49), K_4_ (122.38), and spermine (649.46) [18,39]. We infer that the high efficiency of charge neutralization is due to the highly positively charged and compact structure of lysozyme. At this critical concentration of lysozyme, the corresponding condensing force reaches its maximal value of 10.7 pN. After the charge inversion, the condensing force decreases gradually due to electrostatic repulsion, and the corresponding morphology of DNA becomes less compact because of the repulsive force.

## Figures and Tables

**Figure 1 polymers-14-01377-f001:**
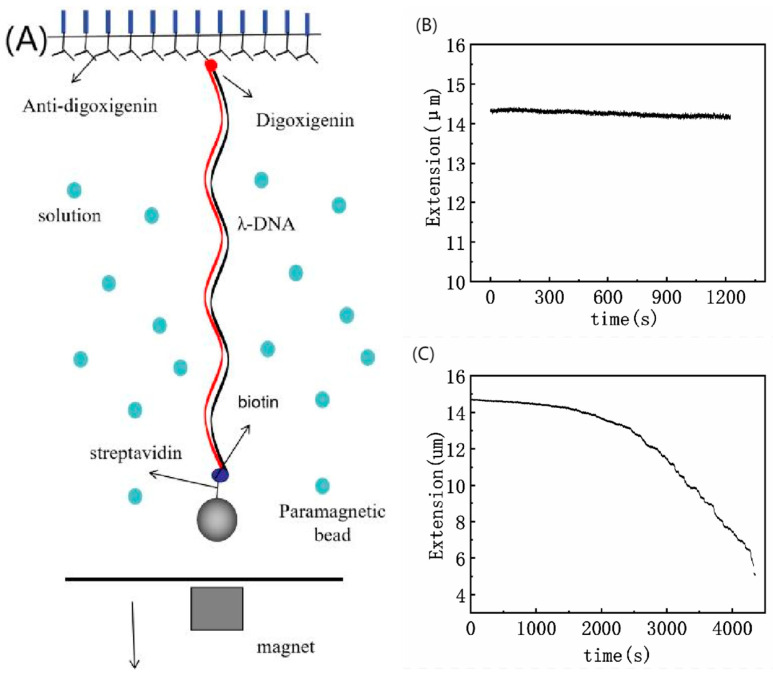
(**A**) A schematic diagram of the experiment setup; The curves of condensing forces are shown in (**B**) a non-condensing curve and (**C**) a condensing curve. DNA extension-time curve measured by MT in DNA compaction process with lysozyme.

**Figure 2 polymers-14-01377-f002:**
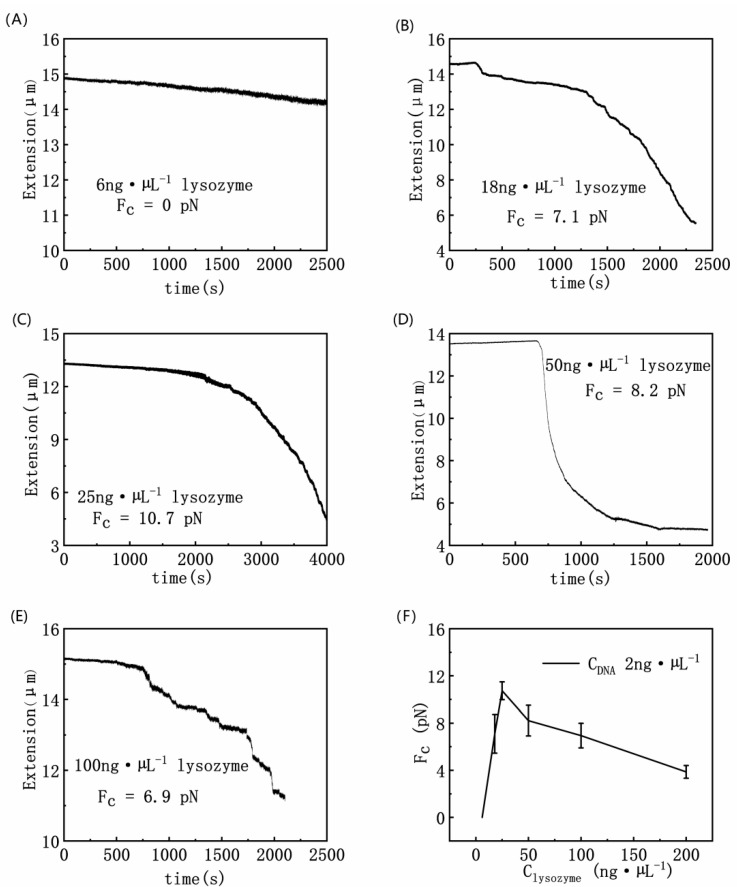
DNA contraction curve as a function of time at different lysozyme concentrations ((**A**) 6 ng·µL^−1^ lysozyme; (**B**) 18 ng·µL^−1^ lysozyme; (**C**) 25 ng·µL^−1^ lysozyme; (**D**) 50 ng·µL^−1^ lysozyme; (**E**) 100 ng·µL^−1^ lysozyme; (**F**) DNA condensing force of DNA–lysozyme complexes as a function of lysozyme concentrations of solutions (F: 6 ng·µL^−1^,18 ng·µL^−1^; 25 ng·µL^−1^, 50 ng·µL^−1^, 100 ng·µL^−1^, 200 ng·µL^−1^).

**Figure 3 polymers-14-01377-f003:**
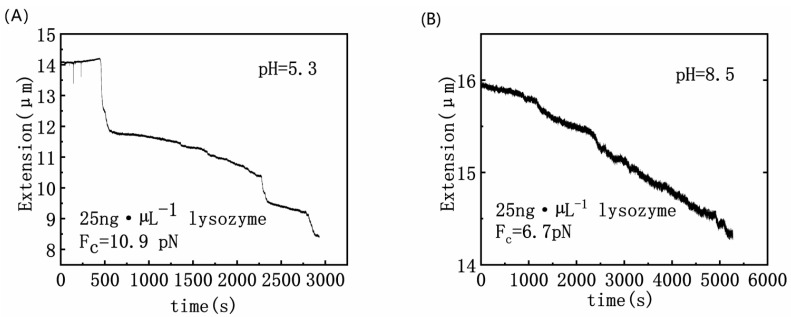
Different pH, DNA contraction curve as a function of time at lysozyme concentration of 25 ng·µL^−1^. (**A**) pH = 5.3; (**B**) pH = 8.5.

**Figure 4 polymers-14-01377-f004:**
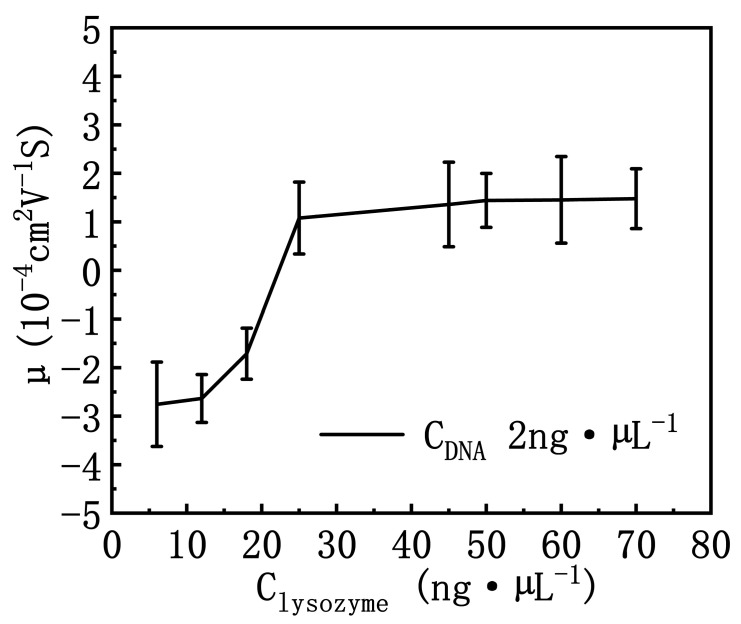
The electrophoretic mobility of DNA measured as a function of lysozyme concentration in a solution of pH = 7. The error bars represent the corrected sample standard deviation.

**Figure 5 polymers-14-01377-f005:**
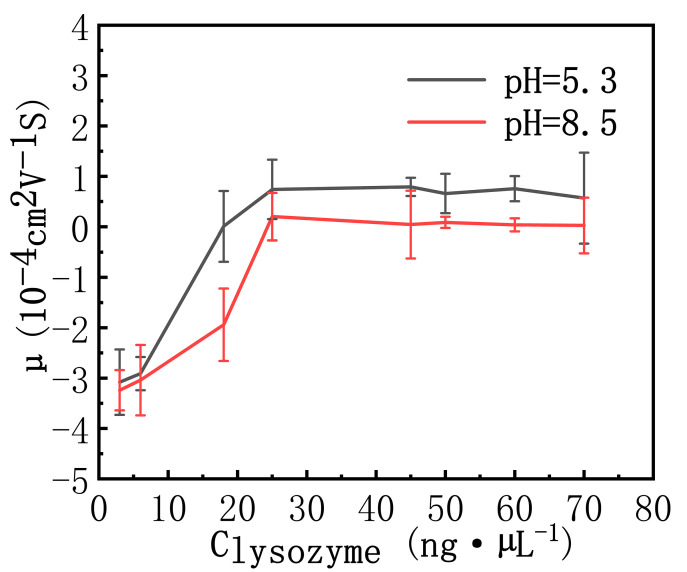
The electrophoretic mobility of DNA measured as a function of lysozyme concentration in a solution of pH = 5.3 and pH = 8.5. The error bars represent the corrected sample standard deviation.

**Figure 6 polymers-14-01377-f006:**
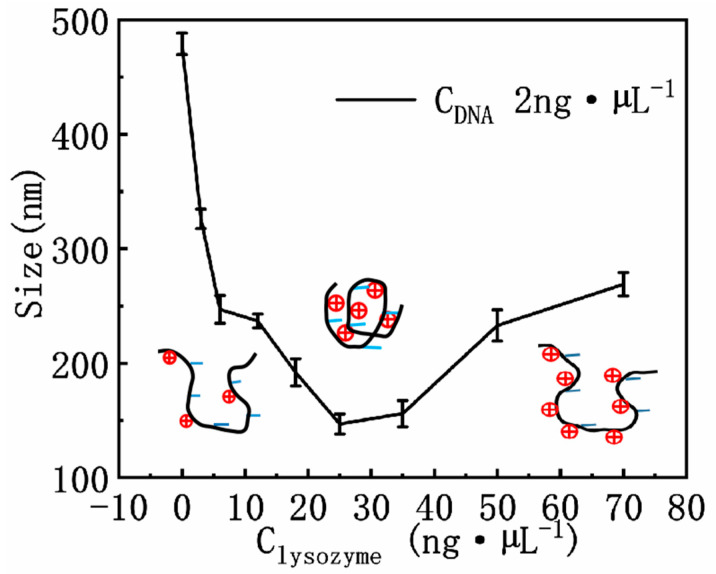
The size of DNA as a function of different lysozyme concentration. The error bars represent the corrected sample standard deviation.

**Figure 7 polymers-14-01377-f007:**
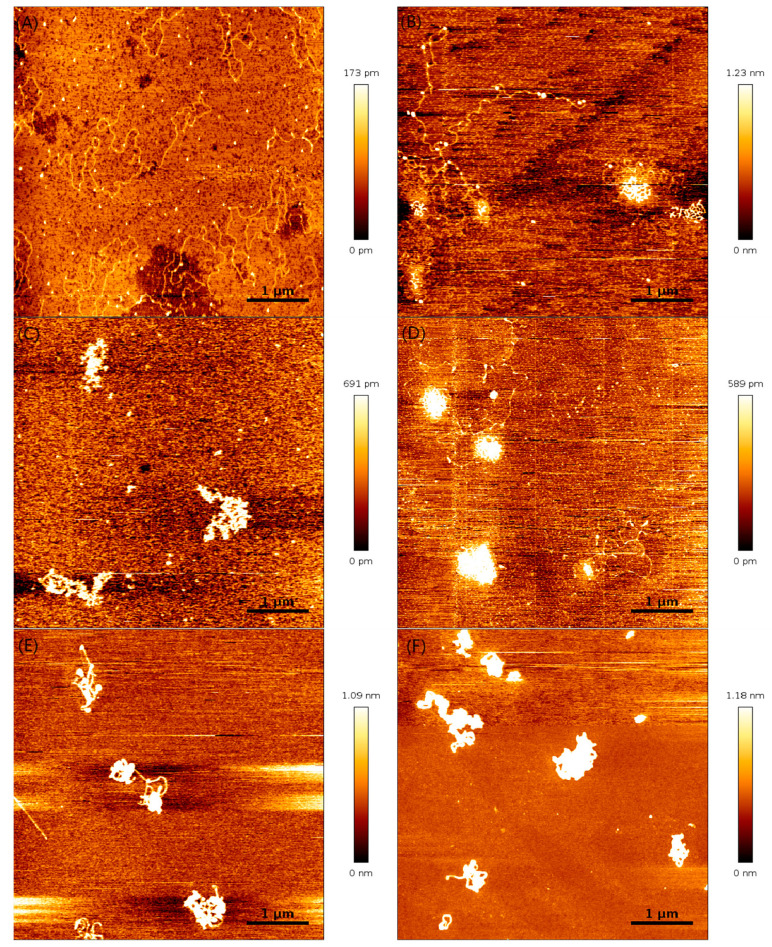
Atomic force images of DNA–lysozyme complexes at different lysozyme concentrations (pH = 7). ((**A**) 0 ng·µL^−1^ lysozyme; (**B**) 6 ng·µL^−1^ lysozyme; (**C**) 10 ng·µL^−1^ lysozyme; (**D**) 18 ng·µL^−1^ lysozyme. (**E**) 35 ng·µL^−1^ lysozyme. (**F**) 50 ng·µL^−1^ lysozyme.).

**Figure 8 polymers-14-01377-f008:**
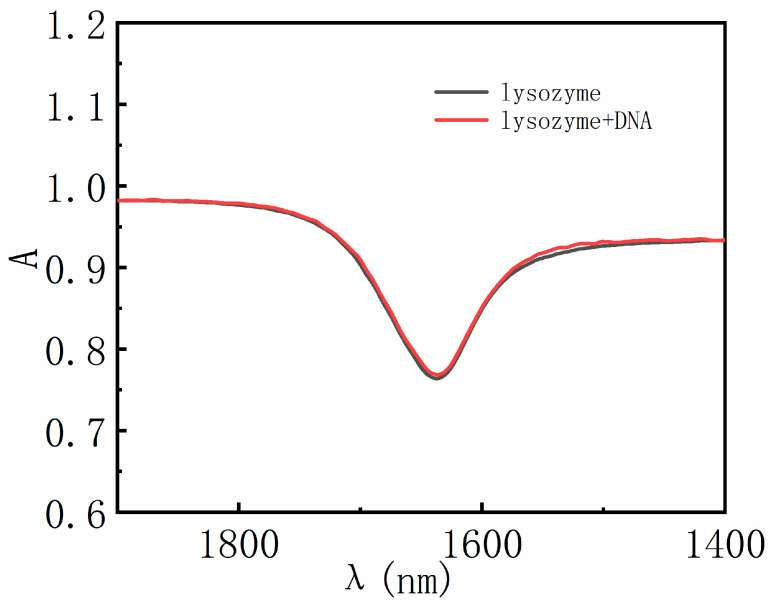
FT-IR spectra of lysozyme and lysozyme–DNA.

## Data Availability

The data presented in this study are available in the article.

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
