# Peer review of "DNA–Lysozyme Nanoarchitectonics: Quantitative Investigation on Charge Inversion and Compaction"

_polymers, 2022, doi:10.3390/polym14071377_

Round 1
Reviewer 1 Report
In my opinion, the work is poorly prepared. Not precisely. The description of the results is modest, no news is visible, no literature review is available, and the introduction does not introduce the topic well. The conclusions do not sum up well. You have to rewrite the whole thing and then rethink the acceptances.
Reviewer 2 Report
The study investigates the DNA/lysozyme interactions by means of magnetic tweezers (MT), atomic force microscopy (AFM) and dynamic light scattering (DLS). The paper is well-structured, it falls within the scope of the journal and is likely to attract the interest of its readers and the wider scientific community. My suggestions/comments are shown below:
- The axes in figures 1 b and 1c are hard to read
- In 2.3 a typo in “minites”
- What is the effect of pH on the DNA/lysozyme interactions?
- How the DLS data were analysed given the form factor of DNA?
- Is it assumed that DNA/lysozyme complexation is spontaneous and irreversible?
- The authors should expand their discussion regarding the origin of the increase in DNA size upon further addition of lysozyme (above 30 ng/μl)
Reviewer 3 Report
This work provides certainly valid quantitative data for DNA and lysozyme interactions. Publication of these data in public journal media would have some contributions to the related research fields. However, presentation ways have to be improved. Some basic data have to be supplied. Several revisions are necessary. Please see below.
1) The presented data suggest that charge neutralization between DNA and lysozyme would be a key to effective compaction. Such mechanisms had better be illustrated as Conclusive figure.
2) The current title gives impression of point-less less-innovative research to this work. Inclusion of a new conceptual term to make the title more attractive is recommended. I may suggest use an emerging conceptual term, nanoarchitectonics (as post-nanotechnology concept, see https://pubs.rsc.org/en/content/articlelanding/2021/NH/D0NH00680G). For example, the title like ... DNA-lysozyme nanoarchitectonics: quantitative investigation on charge inversion and compaction ... may sound more attractive.
3) Chemical structure evaluations to investigate interaction between DNA and lysozyme had better be done more by appropriate spectral methods such as FT-IR measurement.
4) Please also add consideration of pH effects. It is sufficiently influential for the target phenomena.
5) Please provide height information for AFM images.
6) Please correct reference styles.
Round 2
Reviewer 1 Report
The improved work meets the standards, although the description of the results should be more carefully compared with the white literature
Reviewer 2 Report
The authors have made a good effort in revising their manuscript
Reviewer 3 Report
Replies and revisions are fine. The revised version becomes acceptable.